

# Evaluation of the correlation of dorsal root ganglia and spinal nerves with clinical symptoms in patients with postherpetic neuralgia using magnetic resonance neurography

Xueqin Cao[1], Bo Jiao[1], Donglin Wen[2], Guangyou Duan[3], Mi Zhang[4], Caixia Zhang[1], Gang Wu[2] and Xianwei Zhang[1]

[1] Department of Anesthesiology, Hubei Key Laboratory of Geriatric Anesthesia and Perioperative Brain Health, and Wuhan Clinical Research Center for Geriatric Anesthesia, Tongji Hospital, Tongji Medical College, Huazhong University of Science and Technology, Wuhan, Hubei, China
[2] Department of Radiology, Tongji Hospital, Tongji Medical College, Huazhong University of Science and Technology, Wuhan, Hubei, China
[3] Department of Anesthesiology, The Second Affiliated Hospital, Chongqing Medical University, Wuhan, Hubei, China
[4] Department of Anesthesiology, Zhongnan Hospital, Wuhan University, Wuhan, Hubei, China

Corresponding authors
Gang Wu,
tongjiwugang1984@qq.com
Xianwei Zhang, ourpain@163.com

## ABSTRACT

**Purpose:** To assess changes of dorsal root ganglia (DRG) and spinal nerves in patients with postherpetic neuralgia (PHN), and investigate the correlation between DRG morphology and clinical symptoms in PHN patients using magnetic resonance neurography (MRN).

**Methods:** In this case-control study, forty-nine lesioned DRG in 30 patients and 49 normal DRG in 30 well-matched (age, sex, height, weight) healthy controls were assessed. Clinical symptoms of patients (pain, allodynia, itching, and numbness) were assessed. MRN features (DRG volume ($V_{DRG}$), the largest diameter ($D_{max}$) of spinal nerves, signal intensity of DRG and spinal nerves (M-value)) were measured in all participants. Multilinear regression analysis was used to evaluate the relationship between the DRG morphology and clinical symptoms in patients.

**Results:** The volume and relative M-value of lesioned DRG in patients were significantly higher than those on the same side of healthy controls ($p = 0.013$, $p < 0.001$, respectively). The mean $D_{max}$ and relative M-value of spinal nerves on the lesioned side were significantly higher than those on the contralateral and same side of healthy controls ($p < 0.0001$, $p = 0.0001$, $p = 0.0011$, $p = 0.0053$, respectively). No difference was found between the mean $V_{DRG}$ of the lesioned and contralateral sides. Multiple linear regression analysis revealed that disease duration was independent risk factor for the maximum rate of $V_{DRG}$ differences ($p = 0.013$).

**Conclusions:** DRG and spinal nerves on the lesioned side are swollen during PHN. Disease duration is an independent risk factor for morphological differences in the lesioned DRG of PHN patients. This study provides important guidance for individualized treatments of PHN.

## INTRODUCTION

Postherpetic neuralgia (PHN), the most common complication of Herpes zoster (HZ), is defined as pain persisting for >1 month after healing of herpetic skin lesions or pain persisting for >3 months following HZ onset (*Johnson & Rice, 2014*). Its duration varies from a few months to a lifetime (*Kim, Kim & Jo, 2017*).

DRG-based targeted therapies, such as radiofrequency modulation and drug injection, are common interventional treatments for PHN. However, their effect varies among individuals, often leading to unsatisfactory treatment results in patients with a prolonged disease duration (*Johnson & Rice, 2014*; *Kim, Kim & Jo, 2017*; *Aggarwal et al., 2020*). It remains unclear, whether the different treatment effects are related to DRG changes, what role the course of the disease (≤3 *vs* >3 months) plays, and whether there is a relationship between the different clinical phenotypes of PHN and DRG lesions.

Since *in vivo* examination of DRG in patients with PHN is challenging, previous studies mainly focused on cadavers and animals. A postmortem analysis of three patients with severe PHN showed a loss of ganglion cells, axons, myelin, and atrophy of the dorsal horn (*Watson et al., 1991*). Inflammatory responses in the spinal dorsal horn were found in another postmortem analysis of one patient with PHN for 5 weeks, manifesting as macrophage and lymphocyte infiltration, vacuolization of the dorsal horn, but without inflammation of the nerve roots (*Moshayedi et al., 2018*; *Karmarkar, 2007*). However, comparing anatomic and clinical studies is challenging because of the difference between cadavers and living humans, as well as delayed autopsies after clinical symptom onset. Understanding dynamic changes in DRG *in vivo via* noninvasive examinations is necessary for the clinical research and treatment of PHN.

Therefore, this study aimed to apply magnetic resonance neurography (MRN) for the evaluation of DRG morphology and inflammatory changes in patients with PHN, along with the morphological and inflammatory differences in the spinal nerves at the peripheral end of the DRG. We expected to (1) confirm that DRG and spinal nerves are critical in the pathogenesis of PHN and (2) clarify the correlation between clinical phenotypes and DRG imaging changes, thereby providing guidance for individualized treatments of PHN patients.

## MATERIALS AND METHODS

This case-control study was conducted in accordance with the Declaration of Helsinki, and the protocol was approved by the Ethics Committee of Tongji Medical College, Huazhong University of Science and Technology, Wuhan, China (2021S084). It is registered at ClinicalTrials.gov. It was written consent given by all subjects for inclusion before they participated in the study.

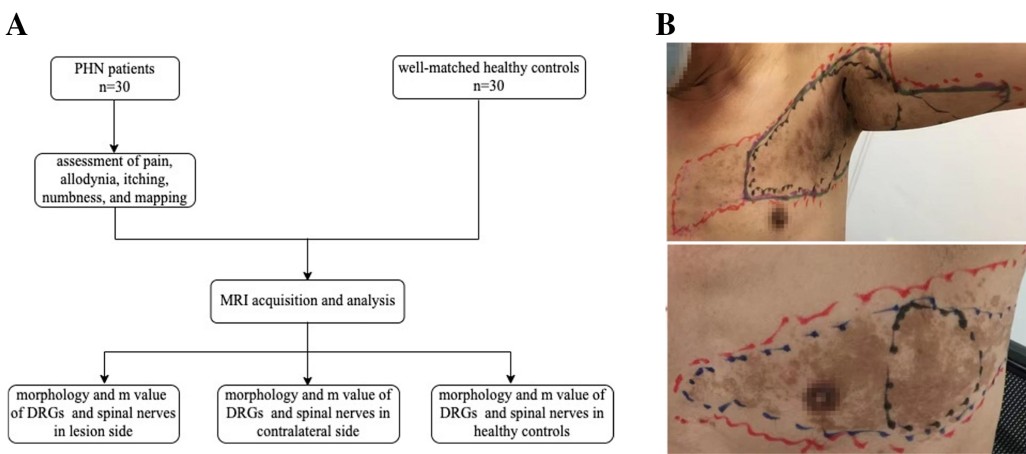

**Figure 1 Study flow chart and skin lesions locating.** (A) Study flow chart. (B) Locating and mapping areas of skin lesions, the most pain area (MPA), allodynia, itching and numbness (red: skin lesions, black: MPA, blue: allodynia, purple: itching, green: numbness).

## Study design and population

This study was designed with a statistical power of 80.14% using GPOWER software3.1 with the following parameters: two-tailed; α = 0.05; and $n = 35$. The study was conducted on 60 individuals. Thirty patients with PHN in the thoracic segments admitted to our hospital between January 2021 and February 2022, and 30 age-, sex-, height-, and weight-matched healthy controls were recruited. The patient inclusion criteria were age >18 years and pain lasting for at least 1 month after lesion crusting. The exclusion criteria were as follows: presence of severe, systemic, metabolic, or neurological diseases that could be correlated with peripheral polyneuropathies, namely multiple myeloma, diabetes mellitus, or thyroid disease; history of psychiatric diseases, other chronic pain, or substance abuse; failure or inability to complete MRN scans; and history of thoracic surgery and pain.

The primary objective was to evaluate differences in DRG volume between the lesioned side of patients and the same side of healthy controls. The secondary outcomes were to examine differences in DRG volume between the lesioned and contralateral sides, differences in the largest diameter ($D_{max}$) of spinal nerves between the lesioned side of patients and the same side of healthy controls, differences in $D_{max}$ of the spinal nerves between the lesioned and contralateral sides, and correlations between clinical symptoms and differences in DRG (Fig. 1A).

Since DRG adjacent to the lesion segments may also be affected (*Kramer et al., 2019*), the volume and M-value of the DRG, as well as the $D_{max}$ and M-values of the spinal nerves of the first segment below or above the lesioned segments, were also evaluated.

## Assessment of pain, allodynia, itching, numbness, and mapping

During the patient's first visit, a numeric rating scale (NRS; 0 = no pain, 10 = worst pain imaginable) was used to evaluate the average daily pain of patients over the past 48 h. Four areas were mapped on the skin: (1) areas of skin lesions; (2) the most painful area (MPA);

(3) areas of dynamic mechanic allodynia (DMA); and (4) areas of sensory disturbance (itching and numbness) (Fig. 1B).

Skin lesion areas were tested with linen tape and estimated using a parallelogram. MPA areas were delineated according to the patients' descriptions. The distribution of DMA was measured by light stroking with a sterile swab from the normal skin towards the most painful area in steps of 1 cm at intervals of 1 s (*Petersen & Rowbotham, 2010*; *Reda et al., 2013*). The areas of itching and numbness were also tested by gently stroking the skin with a sterile swab in steps of 1 cm at intervals of 1 s or according to the patients' descriptions. The intensity of itching was rated on a 0–10 NRS (0 = no itching, 10 = most itching imaginable). All markings were captured as photographs after obtaining patients' consent.

## Locating lesion dermatomes

The lesion dermatomes of all patients were located according to the schematic of cutaneous nerve distribution in the trunk (*Netter, 2003*) by two pain specialists with ≥10 years of experience (Xueqin Cao and Xianwei Zhang).

## Imaging protocol

All patients underwent MRI scans before the start of treatment. We used a 3.0 Tesla MRI scanner (Magnetom Skyra, Siemens). All participants laid supine on the scanner bed with their thorax immobilized in a tight-fitting thoracic coil, and underwent MRN as follows: Three-dimensional T2-weighted sampling perfection with application-optimized contrasts using different flip-angle evolution short-tau inversion-recovery sequence of the spine was used for DRG and spinal nerve imaging: coronal plane, repetition time/echo time 3,000/178 ms, inversion time 220 ms, the field of view $305 \times 240$ mm$^2$, matrix size $320 \times 320$, slice thickness 1.0 mm, slice number 60; no gap, voxel size $1.0 \times 1.0 \times 1.0$ mm$^3$, 50% phase over-sampling, and acquisition time 8 min 35 s; imaging the thoracic spine included all bilateral DRG and spinal nerves from the third upper to the third lower segment centered on the lesioned DRG and spinal nerves corresponding to the damaged dermatomes. All bilateral DRG and spinal nerves of the same segments were scanned in the healthy controls.

## Image analysis

The DRG volumes were assessed using 3D-slicer version 5.0.3 (http://www.slicer.org/) and defined as the $V_{DRG}$. Signal intensity in T2-weighted imaging (M-value) was measured to indirectly reflect inflammation of the DRG and spinal nerves. The Synapse 3D software package (version 5.0.3) was used to measure the $D_{max}$ of the spinal nerves and the M-values of the DRG and spinal nerves. Measurements of the M-value of the DRG were performed in the selected slice with the largest cross-sectional area. $D_{max}$ and M-values of the spinal nerves were measured in the selected slice with the largest diameter (Fig. 2). The imaging data were measured and then re-measured after 2 weeks by a radiologist (Donglin Wen) to assess intra-observer reliability. Another radiologist (Gang Wu) independently performed the measurements to assess inter-observer reliability. Both radiologists were blinded to the clinical data. Intra- and inter-observer reliability were evaluated through correlation coefficients (ICCs). ICCs > 0.75 indicated good agreement.

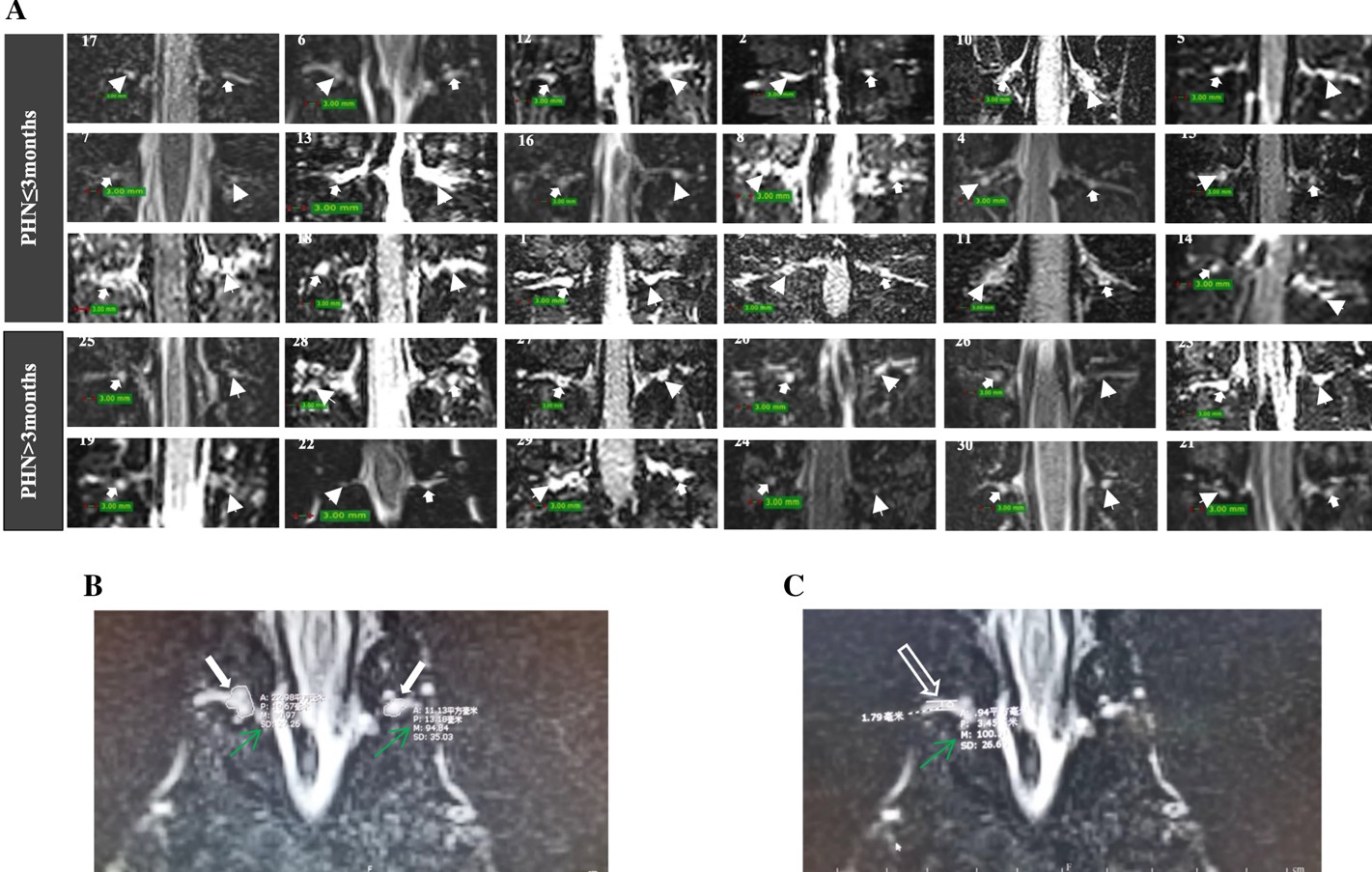

**Figure 2 MRI of DRG and spinal nerve of patients with PHN.** (A) The DRG morphology in lesioned and contralateral side in patients with disease duration ≤3 or >3 months (arrowhead: lesioned side; solid arrow: contralateral side; patient number was showed in each image). (B) The largest cross-sectional area of DRGs were selected (solid arrow), the M value of DRG was measured (green arrow). (C) The largest diameter (hollow arrow) and M value of spinal nerve (green arrow) was measured.

Considering the influence of height and weight on the morphology of the DRG and spinal nerves, the maximum rate of $V_{DRG}$ difference ($\Delta V_{DRG} = V_{DRG}$ (lesioned side)$-V_{DRG}$ (contralateral side)/$V_{DRG}$ (contralateral side)) was used to evaluate the correlation between $V_{DRG}$ differences and clinical symptoms. The signal intensity on T2-weighted imaging of each patient's normal disc was measured as a reference, and the relative M-values of the DRG and spinal nerves ($M_{DRG}/M_{disc}$ and $M_{nerve}/M_{disc}$, respectively) were used for assessment.

## Treatment and follow-up

The patients' treatment plans were solely formulated according to the patients' clinical conditions. Patients were followed up *via* phone at 1-, 3-, and 6-months post-treatment to assess NRS ratings for pain and itching intensities.

## Statistical analysis

Data analyses were performed using SPSS 22 software (SPSS Inc., Chicago, IL, USA). The normality of continuous variables was confirmed with Shapiro–Wilk tests, and the homogeneity of variance was evaluated using Levene's test. Normally distributed data were described as $\bar{\chi} \pm SD$. The data were compared between the two groups using paired-samples or independent-samples $t$-tests. Non-normal data were described as the median (interquartile range), and between-group differences were examined using the Wilcoxon or Mann–Whitney U tests. Correlations among normally distributed continuous data were analyzed using Pearson's correlation coefficient and simple linear regression. Non-normal continuous data and categorical variables were evaluated using Spearman's correlation and the Chi-square test. Statistically significant variables were selected for multilinear regression analysis. The assumption of independence for the residuals was assessed using the Durbin–Watson test. The assumptions of normality and homogeneity of variance for residuals were assessed using histograms and residual plots, respectively. Multicollinearity was assessed using Pearson's correlation coefficient statistics and by checking the variance inflation factor for the same dependent and independent variables. All statistical tests were two-sided, and statistical significance was set at $p < 0.05$.

# RESULTS

## Characteristics of patients and healthy controls

The mean age of the patients was $60.7 \pm 10.18$ years. Forty-nine clinical lesion dermatomes were identified in 30 patients. Age, sex, height, and weight did not differ between the patient and healthy control groups (all $p > 0.050$). The demographic and clinical characteristics of the two groups are summarized in Tables 1 and 2, respectively. Intra-and inter observer reliability was evaluated through correlation coefficients and ICCs were 0.91 and 0.80, respectively, indicating good stability.

## Comparison of DRG and spinal nerves between the lesioned side of patients and the same side of healthy controls

The mean $V_{DRG}$ and relative M-value of DRG were significantly higher in the lesioned side of patients than in the same side of healthy controls ($p = 0.013$, $p < 0.001$, respectively). The mean $D_{max}$ and relative M-value of the spinal nerves were significantly higher on the lesioned side of patients than on the same side of healthy controls ($p < 0.001$, $p < 0.001$, respectively) (Fig. 3).

## Comparison of DRG and spinal nerves between the lesioned and contralateral sides in patients

No difference was found in the mean $V_{DRG}$ and relative M-value of DRG between the lesioned and contralateral sides ($p = 0.089$, $p = 0.715$, respectively). The mean $D_{max}$ and relative M-value of the spinal nerves were significantly higher on the lesioned side than on the contralateral side ($p = 0.001$, $p = 0.005$, respectively) (Fig. 4).

**Table 1 Characteristics of postherpetic neuralgia (PHN) patients and healthy control group.**

| Characteristic | PHN cases (*n* = 30) | Healthy controls (*n* = 30) | *p* |
|---|---|---|---|
| Mean age, years (per patient) | 60.70 ± 10.18 | 58.13 ± 10.54 | 0.341 |
| Sex, n; female/male | 12/18 | 12/18 | |
| Mean height-cm (per patient) | 163.4 ± 7.29 | 164.0 ± 7.22 | 0.598 |
| Mean weight-kg (per patient) | 61.1 ± 11.45 | 62.9 ± 7.91 | 0.468 |
| Median disease durations (IQR)—months (per patient) | 2.65 (1.5–6.0) | — | |
| 1 month<DD≤3 months | 18 (60%) | | |
| DD>3 months | 12 (40%) | | |
| Pain intensity— (NRS 010) (per patient) | 6.9 ± 1.5 | — | |
| Total lesion dermatomes | 49 | — | |
| 1 month<DD≤3 months | 31 | | |
| DD>3 months | 18 | | |
| Mean lesion dermatome (per patient) | 1.63 ± 0.49 | — | |
| Location of skin lesion | 18 (60%) | — | |
| Left | 12 (40%) | | |
| Right | | | |
| Allodynia | 27 (90%) | — | |
| Numbness | 11 (36.67%) | — | |
| Itching | 15 (50%) | — | |

**Note:**
Continuous data are presented as means ± SDs or mean (IQR); data in parentheses are interquartile ranges. Categorical data are numerators; data in parentheses are percentages. DD, disease duration.

## Correlation of $\Delta V_{DRG}$ with age, sex, disease duration, and skin lesion area

Age and sex were not correlated with $\Delta V_{DRG}$ ($p = 0.738$ and $p = 0.456$, respectively). Disease duration and skin lesion area were significantly negatively correlated with $\Delta V_{DRG}$ ($r = -0.435$, $p = 0.003$; $r = -0.540$, $p = 0.002$, respectively).

## Correlation of $\Delta V_{DRG}$ with allodynia, numbness, itching, pain severity, and itching severity

Itching and $\Delta V_{DRG}$ were significantly positively correlated ($r = 0.443$, $p = 0.014$). The NRS values for pain and itching severity were not significantly associated with $\Delta V_{DRG}$ ($p = 0.929$ and $p = 0.228$, respectively). Additionally, allodynia ($p = 0.360$) and numbness ($p > 0.999$) were not significantly correlated with $\Delta V_{DRG}$.

## Effect of disease duration, area of skin lesion, and itching on $\Delta V_{DRG}$

Multiple linear regression analysis showed that itching and disease duration were independent risk factors for $\Delta V_{DRG}$ ($\beta = 0.376$, $p = 0.017$; $\beta = -0.388$, $p = 0.013$, respectively). Skin lesion area was not significant independent risk factor for $\Delta V_{DRG}$ ($\beta = -0.163$, $p = 0.308$) (Table 3).

**Table 2 Clinical characteristics of each patient.**

| Patients | Age(y)/sex | Location of lesion | Disease duration (months) | Pain intensity (NRS 0–10) | Itching (yes or no) | Itching intensity (NRS 0–10) | Numbness (yes or no) | Allodynia (yes or no) |
|---|---|---|---|---|---|---|---|---|
| 1 | 70/M | L/T7–T8 | 1.5 | 8 | Yes | 7 | No | Yes |
| 2 | 49/M | R/T6–T7 | 2.5 | 5 | Yes | 3 | Yes | Yes |
| 3 | 67/F | L/T8 | 1.4 | 5 | No | 0 | No | Yes |
| 4 | 66/M | R/T11–T12 | 1.6 | 6 | No | 0 | Yes | Yes |
| 5 | 57/M | L/T2–T3 | 1.5 | 8 | Yes | 2 | Yes | Yes |
| 6 | 47/F | R/T5–T6 | 1.5 | 9 | No | 0 | Yes | Yes |
| 7 | 57/M | L/T11–T12 | 1.5 | 9 | Yes | 3 | No | Yes |
| 8 | 63/F | R/T7–T8 | 1.5 | 6 | Yes | 2 | No | Yes |
| 9 | 70/M | R/T4 | 1.7 | 6 | Yes | 2 | Yes | Yes |
| 10 | 53/M | L/T11–T12 | 1.5 | 6 | Yes | 2 | No | No |
| 11 | 72/M | R/T11–T12 | 1.5 | 9 | No | 0 | No | Yes |
| 12 | 48/F | L/T6–T7 | 1.5 | 8 | No | 0 | No | Yes |
| 13 | 59/F | L/T6 | 1.5 | 9 | No | 0 | Yes | No |
| 14 | 82/F | L/T6 | 2 | 7 | No | 0 | No | No |
| 15 | 67/M | R/T2–T3 | 2 | 5 | Yes | 2 | No | Yes |
| 16 | 61/M | L/T3 | 2.8 | 9 | No | 0 | No | Yes |
| 17 | 36/M | R/T2–T3 | 2.9 | 7 | No | 0 | No | Yes |
| 18 | 69/F | L/T8–T9 | 2.9 | 5 | No | 0 | No | Yes |
| 19 | 61/M | L/T2–T3 | 3.1 | 9 | Yes | 2 | Yes | Yes |
| 20 | 53/F | L/T3 | 3.5 | 7 | Yes | 3 | No | Yes |
| 21 | 79/F | R/T4–T5 | 4.5 | 8 | Yes | 3 | Yes | Yes |
| 22 | 65/F | R/T9–T10 | 4 | 8 | Yes | 3 | Yes | Yes |
| 23 | 61/M | L/T4 | 7 | 8 | No | 0 | Yes | Yes |
| 24 | 67/M | L/T3 | 6.5 | 6 | No | 0 | No | Yes |
| 25 | 47/M | L/T2 | 6 | 6 | Yes | 5 | No | Yes |
| 26 | 59/M | L/T2–T3 | 7.5 | 7 | Yes | 6 | No | Yes |
| 27 | 52/M | L/T8–T9 | 9.5 | 5 | No | 0 | No | Yes |
| 28 | 51/F | R/T11 | 15 | 7 | Yes | 3 | Yes | Yes |
| 29 | 65/M | R/T5 | 18 | 4 | No | 0 | No | Yes |
| 30 | 68/F | L/T2–T3 | 42 | 6 | No | 0 | No | Yes |

**Note:**

M, male; F, female; T, level of thoracic vertebrae; L, left side of thoracic vertebrae; R, right side of thoracic vertebrae.

## Difference in morphology and M-value of bilateral DRG and spinal nerves in the first segment adjacent to lesion segments

No significant difference was found in DRG or spinal nerves between the two sides in the first segment adjacent to lesion segments ($p > 0.05$). Also, no significant difference was found in DRG or spinal nerves between the two sides in the third segment away from the lesion segments ($p > 0.05$).

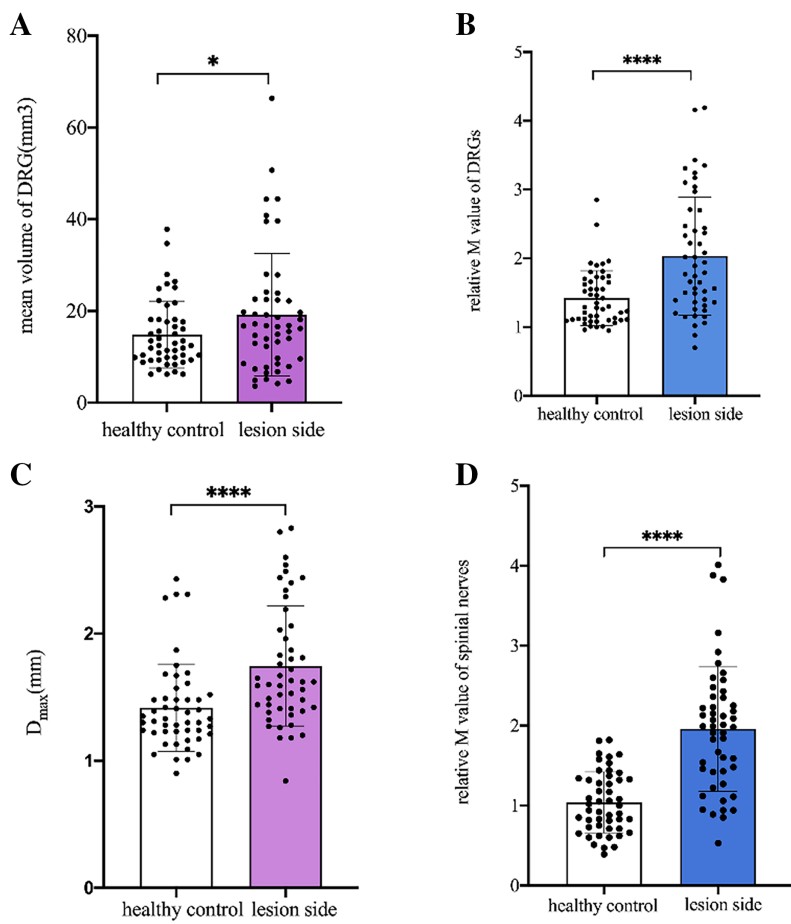

**Figure 3 Comparison of DRGs and spinal nerves between lesioned side of patients and healthy controls.** (A) The mean $V_{DRG}$ of DRGs in the lesioned side of patients were significantly more than that in the same side of healthy controls (19.2 ± 13.3 $vs$ 14.85 ± 7.308, $^*p$ = 0.013). (B) The relative M value of DRGs in the lesioned side of patients were significantly more than that in the same side of healthy controls (2.03 ± 0.86 $vs$ 1.42 ± 0.40, $^{****}p < 0.001$). (C) The mean Dmax of spinal nerves in the lesioned side of patients was significantly more than that in the same side of healthy controls (1.746 ± 0.47 $vs$ 1.418 ± 0.34, $^{****}p < 0.001$). (D) The relative M value of spinal nerves in the lesioned side of patients was significantly more than that in the same side of healthy controls (1.96 ± 0.78 $vs$ 1.04 ± 0.38, $^{****}p < 0.001$). Paired-samples t-tests was used to compare the data between two groups.

## NRS score of pain and itching during follow-up

No significant correlations were found between $\Delta V_{DRG}$ and itching severity at 1-, 3-, and 6-months post-treatment ($p$ = 0.097, $p$ = 0.808, and $p$ = 0.295, respectively). Similar results were observed for the correlations between $\Delta V_{DRG}$ and pain severity at 1-, 3-, and 6-months post-treatment ($p$ = 0.124, $p$ = 0.790, and $p$ = 0.295, respectively). However, patients with positive $\Delta V_{DRG}$ recovered better than those with negative $\Delta V_{DRG}$.

At 3 months post-treatment, all patients reported an NRS score of pain and itching of <5. Among the 12 patients with negative $\Delta V_{DRG}$, the incidence of NRS >3 was 25% (pain) and 8.3% (itching); this result was 16.7% (pain) and 0% (itching) in the 18 patients with positive $\Delta V_{DRG}$. At 6 months post-treatment, the NRS scores for pain and itching were <5.

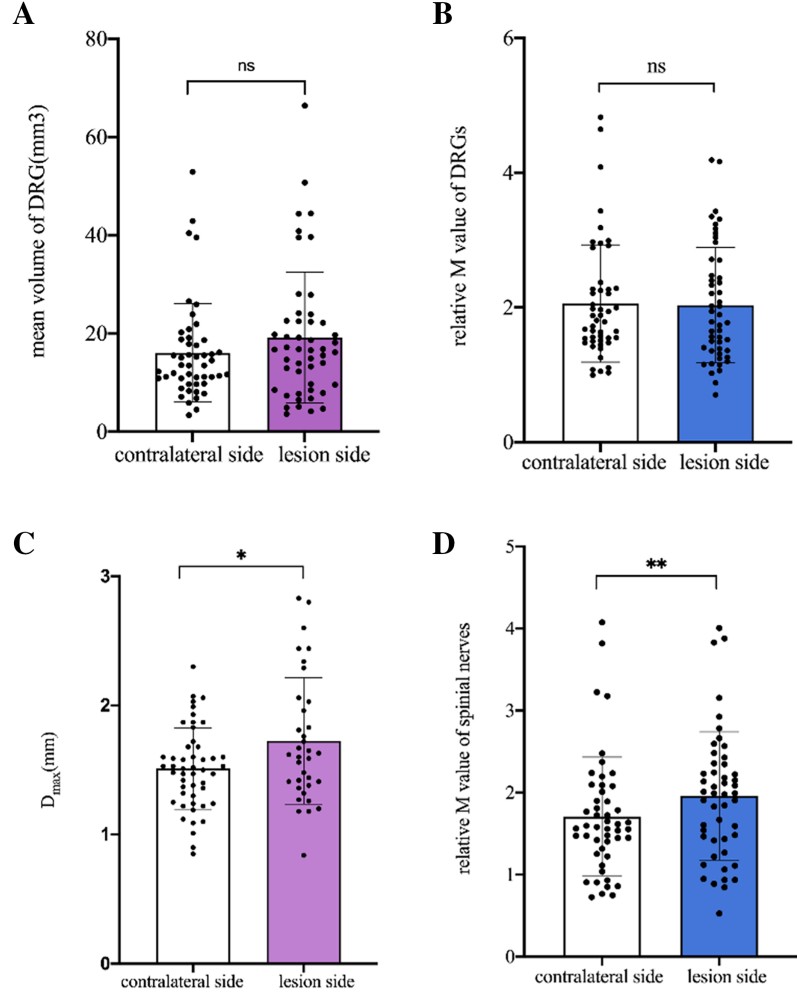

**Figure 4 Comparison of DRGs and spinal nerves between lesion and contralateral side in patients.**
(A) No difference was found in the mean $V_{DRG}$ of DRG between the lesion and contralateral sides (19.2 ± 13.32 *vs* 16.05 ± 0.99, *p* = 0.089). (B) No difference was found in the relative M-value of DRG between the lesion and contralateral sides (2.03 ± 0.86 *vs* 2.06 ± 0.88, *p* = 0.715). (C) The mean Dmax of the spinal nerves was significantly higher on the lesioned side than on the contralateral side (1.746 ± 0.47 *vs* 1.511 ± 0.32, \**p* = 0.001). (D) The relative M-value of the spinal nerves was significantly higher on the lesioned side than on the contralateral side (1.958 ± 0.78 *vs* 1.709 ± 0.72, \*\**p* = 0.005). Paired-samples t-tests was used to compare the data between two groups.

Among the 12 patients with negative $\Delta V_{DRG}$, the incidence of NRS >3 was 41.7% (pain) and 16.7% (itching); this result was 0% (pain) and 0% (itching) in the 18 patients with positive $\Delta V_{DRG}$.

## DISCUSSION

Advanced neuroimaging approaches have been dominating studies on PHN-related brain regions and pain modulation pathways (*Davis et al., 2017*; *Gustin et al., 2011*). Pathological changes in the brain structure and function in patients with PHN and deficits in ascending and descending pain modulation pathways were found (*Chen et al., 2017*; *Li et al., 2020*;

**Table 3 Effect of disease duration, area of skin lesion, and itching on $\Delta V_{DRG}$.** The results showed that the regression equation was significant (F = 6.508, \*\*\*$p$ < 0.001). The disease duration (β = −0.388, $p$ = 0.013) was a significant negative predictor of $\Delta V_{DRG}$, and the occurrence of pruritus (β = 0.376, $p$ = 0.017) was a significant positive predictor of $\Delta V_{DRG}$. Area of skin lesion could not predict $\Delta V_{DRG}$. Together, these variables explain 43.2% of the variation in $\Delta V_{DRG}$.

| Independent variable | B | β | t | $p$ | F | Adjusted $R^2$ |
|---|---|---|---|---|---|---|
| Disease duration | −0.381 | −0.388 | −2.685 | 0.013 | 6.508\*\*\* | 0.432 |
| Area of skin lesion | −6.838 | −0.163 | −1.040 | 0.308 | | |
| Appearance of itching | 0.361 | 0.376 | 2.573 | 0.017 | | |

**Note:**
B, unstandardized regression coefficient; β, standardized regression coefficient.

*Cao et al., 2017*). Even though it is a major site of neuropathic pain signal transmission and maintenance, neuroimaging studies on DRG pathophysiology in PHN are lacking.

In this study, the $V_{DRG}$ and relative M-values of the DRG were significantly larger on the lesioned side of patients than on the same side of healthy controls; this indicated that the DRG on the lesioned side was swollen. However, no difference was found between the mean $V_{DRG}$ and relative M-values of the DRG on the lesioned and contralateral sides. These paradoxical results imply that the DRG on the contralateral side may have been swollen simultaneously.

Furthermore, multiple linear regression analysis showed that disease duration was negatively related to and an independent risk factor for $\Delta V_{DRG}$. This finding could be attributed to the stimulation of the DRG being ineffective in some PHN cases presenting with a prolonged disease duration. The mechanism underlying this change should be further investigated.

The mean $D_{max}$ and relative M-value of the spinal nerves were significantly higher in the lesioned side of patients than in the same side of healthy controls. Meanwhile, the mean $D_{max}$ and relative M-value of the spinal nerves were significantly higher in the lesioned than in the contralateral side. This suggests that the spinal nerves in the lesioned side were significant swollen during PHN, which may be the pathological mechanism of intercostal nerve block in patients with PHN.

The prevalence of postherpetic itch (PHI) in patients with PHN is high, ranging from 30% to 58% (*van Wijck & Aerssens, 2017*; *Oaklander et al., 2003*). The mechanism of PHI remains unclear, and itching may occur transiently at the time of rash healing or intensify as the pain subsides or persists for a long time as a sole symptom without adequate treatment. Preherpetic itching is correlated with PHN (*Kramer et al., 2019*; *Koshy et al., 2018*). In this study, multiple linear regression analysis showed that itching was an independent influencing factor for $\Delta V_{DRG}$. However, no significant correlation was observed between itch intensity and $V_{DRG}$. These results suggest that the early onset of itching in patients with PHN might imply that it is difficult to treat and requires aggressive treatment.

Older age, female sex, and high pain intensity are regarded as critical predictors of PHN (*Forbes et al., 2016b, 2016a*). However, no significant correlation was observed between these factors and morphological features of the DRG or spinal nerves in this study,

suggesting that they may not play an obvious role in the morphological changes in DRG and spinal nerves in patients with PHN. Previous research has suggested that the area of the skin lesion is a predictor of PHN (*Wang, Zhang & Fan, 2020*). In this study, although the skin lesion area correlated with $\Delta V_{DRG}$, it was not an independent risk factor for $\Delta V_{DRG}$. Furthermore, numbness and allodynia did not correlate with the morphological features of the DRG.

However, various mechanisms are involved in the process of PHN. According of many recent reports, pain (in its various forms, such as spontaneous pain, tactile allodynia *etc.*,) and itch are both presumably related to abnormal impulse initiation caused by resurgence of infection with the zoster virus. This begins in the DRG and then migrates to the skin (HZ), and amplified by central sensitization. But if the immune response is inadequate the virus goes on to sicken and eventually kill DRG neurons (*Devor, 2018*). The present study solely demonstrates the macroscopic morphological and signal intensity alterations of the DRG, while the correlation between microstructural changes and clinical symptoms necessitates additional investigation. Furthermore, the spinal dorsal horn and central mechanisms also play critical roles in PHN (*Watson et al., 1991*; *Cao et al., 2017*; *Tansley et al., 2022*). Whether an association exists between clinical symptoms and these mechanisms requires further studies.

This study has two limitations. First, the sample size was limited, which impedes the mechanistic investigation of PHN. Second, patients with pain for less than 3 months are part of the cohort (*Li et al., 2020*; *Cao et al., 2017*). Some studies have defined PHN as pain persisting for at least 3 months after the appearance of a rash (*Johnson & Rice, 2014*). Whether the current findings can be generalized to patients with PHN reporting pain beyond 3 months after rash onset should be verified in future studies.

## CONCLUSIONS

In summary, we found that DRG and spinal nerves in the lesioned side were swollen significantly during PHN. Disease duration was an independent risk factor for morphological differences in the DRG on the lesioned side in patients with PHN. Moreover, itching might be an independent risk factor for morphological differences in the DRG on the lesioned side. These results provide critical guidance for individualized treatments of PHN.

### Funding
The authors received no funding for this work.

### Competing Interests
The authors declare that they have no competing interests.

### Author Contributions
- Xueqin Cao conceived and designed the experiments, performed the experiments, analyzed the data, prepared figures and/or tables, and approved the final draft.

- Bo Jiao performed the experiments, authored or reviewed drafts of the article, and approved the final draft.
- Donglin Wen performed the experiments, prepared figures and/or tables, and approved the final draft.
- Guangyou Duan conceived and designed the experiments, authored or reviewed drafts of the article, and approved the final draft.
- Mi Zhang performed the experiments, authored or reviewed drafts of the article, and approved the final draft.
- Caixia Zhang analyzed the data, authored or reviewed drafts of the article, and approved the final draft.
- Gang Wu performed the experiments, analyzed the data, prepared figures and/or tables, and approved the final draft.
- Xianwei Zhang conceived and designed the experiments, authored or reviewed drafts of the article, and approved the final draft.

## Human Ethics

The following information was supplied relating to ethical approvals (*i.e.*, approving body and any reference numbers):

This study was approved by the Ethics Committee of Tongji Medical College, Huazhong University of Science and Technology, Wuhan, China (Ethical Application Ref: 2021S084).

## Data Availability

The raw data are available in the Supplemental File.

## Supplemental Information

Supplemental information for this article can be found online at http://dx.doi.org/10.7717/peerj.15998#supplemental-information.

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
