# Peer review of "Evaluation of the correlation of dorsal root ganglia and spinal nerves with clinical symptoms in patients with postherpetic neuralgia using magnetic resonance neurography"

_PeerJ, doi:10.7717/peerj.15998_

## Round 0.1 · original submission · Minor Revisions

It is my opinion as the Academic Editor for your article - Evaluation of the correlation of dorsal root ganglia and spinal nerves with clinical symptoms in patients with postherpetic neuralgia using magnetic resonance neurography - that it requires a number of Minor Revisions.

·

Basic reporting

No comment

Experimental design

1. Even though it is mentioned in the discussion, the rationale for not following the definition of chronic pain as pain for longer than 3 month is not clear.
If patients with pain for less than 3 month are part of the cohort, then the investigation does not cover chronic pain. Please discuss.

Methods:
2. e.g. lane 111 authors have not been named. The content of the brackets "(Author X and Author Y)" might have to be changed throughout the manuscript.

Validity of the findings

Results:
3. Lane 188 "..Disease course and skin lesion area were negatively correlated with the volume of the DRG area. Is it correct that the larger the skin lesion area was, the smaller the deltaVDRG was? Please clarify.

Discussion:
4. Lane 247-249 "However, no significant correlation was observed...." Even though the authors state that the factors have no impact on the morphology of DRG and spinal nerves, the sentence might be misread as there are plenty of other factors outside oedema that are involved in the pathophysiology of pain and itch. The authors might include the fact that the study is limited to morphological criteria (with the advantage of longitudinal measurements) into the discussion.

Tables:
5. Please check the headings of tables so that all headings start with capital letters.

Figures:
6. Please add scale bars to the images 2a-c

Additional comments

A well conducted and well written investigation

Reviewer 2 ·

Basic reporting

This is a carefully done, and nicely written morphometric analysis of thoracic DRGs and spinal nerves in patients with PHN, based on quantitative MRI. PHN is a common and devastating chronic pain condition that is almost untreatable. However, it is not lethal, or treated by DRG excision, so very little pathological information is available, especially at short times (weeks-months) after disease onset. Although the results pose more questions than they answer, this is a useful contribution to the literature.

I have a number of relatively minor points that the authors may want to consider in a revision.

1) The size parameters used are understandable, but I (and presumably many readers) am not familiar with the significance of M-value, "the signal intensity of DRG and spinal nerves. An increase in imaged volume indicates swelling, and perhaps inflammation. But what is the functional meaning of having a high or low M-value?
2) Abstract, line 79, Table 1 and elsewhere: If you mean biological sex please use "sex", not "gender", which is a psychological construct.
3) Line 83: lists "chronic pain" as an exclusion factor even though it is the principle symptom of your subjects. Explain what you mean… no migraine, no heartburn etc.?
4) Lines 120-122: Did you image the (presumably) infected DRG +/- 3 segments (total 7 DRGs bilaterally), or a total of 6 ?
5) Lines 132-134. Blind evaluation of the images is essential. I don't fully understand the explanation. Author Z measured once, then again 2 weeks later? Did author XY measure a 3rd time, just check the numbers of Author Z, or something else ?
6) Line 167-8: Please explain 49 lesions in 30 patients. Did 19 patients have lesions in more than one dermatome?
7) Line 211: Patients with DRG swelling "recovered better" than those with DRG shrinking (negative volume change). What does "recovered better" mean (resolution of pain and itch?) and do you have an explanation of this? If swelling correlates with pain relief, it probably doesn't indicate inflammation, or maybe it does (see below).
8) Line 205-6: Swelling of the infected DRG and the neighboring one was no different (also on the contralateral side). What about the DRGs +/- 3 segments away from the infected DRG?
9) Figs. and Tables: Several times different terms are used that (apparently) mean the same thing, e.g. "disease duration" and "disease course". Please define each and point out the difference. If they are the same, chose one of them.

10) Discussion: The results on pathogenesis are indeed paradoxical, but you are not alone. In sciatica, osteoarthritis and other painful conditions imaging tends not to correlate with pain and related unpleasant sensations like itch. At least you are looking at neural tissue and not bones. I would recommend that you devote a part of your Discussion to the primary cause of the symptoms measured, pain (in its various forms… ongoing pain, tactile allodynia etc.) and itch. Both are presumably related to abnormal impulse initiation caused by resurgence of infection with the zoster virus. This begins in the DRG and then migrates to the skin (HZ). But if the immune response is inadequate the virus goes on to sicken and eventually kill DRG neurons. The histology available (mostly Watson) shows atrophied DRGs (and dorsal horn), but that's years after the HZ, not a month or three. The swelling you see at these early times (and on both sides, but +/- how many segments?) could well be real and reflect a systemic immune response. Recall that the original varicella zoster infection, decades earlier in childhood, was systemic and probably infected all DRGs, bilaterally. The fact that in HZ and PHN only one DRG seems to be symptomatic probably means that a broad systemic immune response was mounted that was quite powerful. This might be what you are seeing. You might want to Google "Devor, pain, postherpetic neuralgia").

Experimental design

OK

Validity of the findings

OK

Additional comments

Comments above

·

Basic reporting

In this manuscript, the authors investigated the correlation between DRG morphology and clinical symptoms in PHN patients using magnetic resonance neurography (MRN). The relationship between DRG morphology and clinical symptoms in PHN patients was assessed by multi-linear regression analysis and determined that the course of the disease is an independent risk factor that affects the maximum rate of VDRG differences. The paper is very well written, and confirm that DRG and spinal nerves are critical in the pathogenesis of PHN , which provides important guidance for individualized treatments of PHN.

Experimental design

no comment

Validity of the findings

Please indicate in detail the statistical methods used in the annotations of Figures 3 and 4.

Additional comments

no comment

---

## Round 0.2 · accepted · Accept

I have assessed the revision myself. The authors responded to the comments effectively. The manuscript is ready for publication.